# Mortality prediction of inpatients with NSTEMI in a premier hospital in China based on stacking model

**Li Wang[1], Yu zhang[1], Feng Ii[2], Caiyun Li[3], Hongzeng Xu[3]\***

1 College of Marine Electrical Engineering, Dalian Maritime University, Dalian, China, 2 School of Information and Electronic Engineering, Zhejiang Gongshang University, Hangzhou, China, 3 Department of Cardiology, The People's Hospital of China Medical University, The People's Hospital of Liaoning Province, Shenyang, China

\* hongzengxu@foxmail.com

**Data Availability Statement:** The data underlying this study cannot be shared publicly due to restrictions imposed by the People's Hospital of Liaoning Province. However, the data can be made available to qualified researchers who meet the

## Abstract

### Background

Acute myocardial infarction (AMI) remains a leading cause of hospitalization and death in China. Accurate mortality prediction of inpatient is crucial for clinical decision-making of non-ST-segment elevation myocardial infarction (NSTEMI) patients.

### Methods

In this study, a total of 3061 patients between January 1, 2017 and December 31, 2022 diagnosed with NSTEMI were enrolled in this study. A new method based on Stacking ensemble model is proposed to predict the in-hospital mortality risk of NSTEMI using clinical data. This method mainly consists of three parts. Firstly, oversampling technique was used to alleviate the class imbalance problem. Secondly, the feature selection method of Recursive Feature Elimination (RFE) was selected for effective feature selection. Finally, a unique double-layer stacking model is designed to improve the performance of the algorithm. Seven classical artificial intelligence methods of Logistic Regression (LR), Decision Tree (DT), Support Vector Machine (SVM), Random Forest (RF), Adaptive Boosting (ADB), Extra Tree (ET), and Gradient Boosting Decision Tree (GBDT) were selected as candidate models for the base model of the first layer of the model, and extreme gradient enhancement (XGBOOST) was selected as the meta-model for the second layer.

### Results

Patient were divided into the surviving group and the death group, and a total of 57 clinical features showed statistically significant for the two groups and finally included in the subsequent model. The results show that the Area Under Curve (AUC) of the Stacking model proposed in this paper is 0.987, which is higher than that of LR (0.934), DT (0.946), SVM (0.942), RF (0.948), ADB (0.949), ET (0.938) and GBDT (0.920). At the same time, the proposed Stacking model has higher performance than each single model in terms of Accuracy, Precision, Recall and F1 evaluation indicators.

necessary criteria for access. Requests for data can be directed to the Ethics Committee of the People's Hospital of Liaoning Province at +86 024 24016585 or via email at lnsrmyyywc@163.com.

**Funding:** This work was supported by the Natural Science Foundation of Liaoning Province (No. 2023-MS-054). The funders had no role in study design, data collection and analysis, decision to publish, or preparation of the manuscript.

**Competing interests:** The authors have declared that no competing interests exist.

**Abbreviations:** The abbreviations are listed in the S1 File.

## Conclusions

The Stacking model proposed in this paper can integrate the advantages of LR, DT, SVM, RF, ADB, ET and GBDT models to achieve better prediction performance. This model can provide valuable insights for physicians to identify high-risk patients more precisely and timely, thereby maximizing the potential for early clinical interventions to reduce the mortality rate.

## Background

Acute myocardial infarction (AMI) is a kind of coronary atherosclerotic heart disease that seriously threatens human health and survival [1]. AMI can usually be divided into non-ST-segment elevation myocardial infarction (NSTEMI) and ST-segment elevation myocardial infarction (STEMI) based on the presence or absence of ST-segment elevation on the electrocardiogram, both of which have higher hospital mortality rates [2, 3]. In clinical practice, NSTEMI patients tend to receive less attention than STEMI patients, but the mortality rate of NSTEMI is not lower than that of STEMI, and NSTEMI accounts for approximately 70% of all myocardial infarction cases. Therefore, it is crucial to accurately predict the mortality of inpatients with NSTEMI, which also provides auxiliary decision-making for the rational allocation of clinical resources and whether to intervene.

Currently, some risk scoring strategies have been developed to assess the risk of death in AMI patients, including the Global Registry of Acute Coronary Event (GRACE) risk score, Thrombolysis in Myocardial Infarction (TIMI) score, Acute Coronary Treatment and Intervention Outcome Network risk score, Canadian ACS risk score, and ProACS risk score [4, 5]. Some risk scoring strategies can also be used to predict the prognosis of AMI patients. These scores are mainly used in patients with STEMI and NSTEMI, but only a small number of Asian patients are included, so these scores are not fully applicable to Asians especially for Chinese.

In recent years, the application of artificial intelligence in the medical field has been increasing. In particular, machine learning (ML), can promote personalized diagnosis and treatment through computer training without explicitly defined rules by humans [6, 7]. For the research on the risk assessment of hospitalization for AMI patients, Konrad Pieszko et al. used ML algorithm to predict the in-hospital mortality of patients with AMI [8]. The study combined three algorithms, namely linear regression, extreme gradient enhancement (XGBOOST), and Dominance-based Rough Set Balanced Rule Ensemble (DRSA-BRE), with clinical and laboratory examination parameters to predict the in-hospital mortality. And the results showed that DRSA-BRE algorithm had the best performance. Changhu Xiao et al. used logistic regression to analyze variable correlations, and used decision trees, naive Bayes, support vector machines, random forests, and gradient boosting methods to predict the prognosis of STEMI patients [9]. Woojoo Lee et al. used regularization logistic regression, random forest, support vector machine and extreme gradient enhancement to predict short-term and long-term mortality of AMI patients [10]. Pontus et al. used the ML model based on logistic regression and artificial neural networks to predict AMI or death in patients with acute chest pain [11]. Khera et al. developed and validated three machine algorithm models to predict adverse outcomes in AMI patients according to patients' medical history and performance characteristics [12]. R. Fu et al. developed a risk score model to predict the risk of in-hospital death in NSTEMI patients based on the Chinese Acute Myocardial Infarction (CAMI) registration i.e., the CAMI-N-STEMI score [13]. On this basis, Chenxi Song et al. further simplified the model so that it can

carry out risk assessment more quickly [14]. Most of these research methods used indicators such as statistical data and individual laboratory test data. In this paper, more laboratory testing indicators are added, as well as the ratio of monocytes to high-density lipoprotein cholesterol, the ratio of neutrophils to lymphocytes, the ratio of monocytes to lymphocytes, the ratio of neutrophils to high-density lipoprotein cholesterol ratio and other indicators to improve the accuracy of the prediction model. In summary, the use of machine learning has improved the management of cardiovascular diseases [15, 16].

To date, there are few studies on risk assessment for NSTEMI patients. This paper proposes a mortality prediction model for inpatients with NSTEMI and further uses advanced machine learning methods to improve the performance of the model. Considering that each ML method may outperform other methods or make errors in different situations, it is natural to integrate multiple ML methods for better predict results. Stacking is a more powerful integration technique which uses the predictions of multiple base learners as a new final result to train new meta-learners [17–19]. In this paper, we try to use Stacking model to predict the mortality of NSTEMI inpatients based on the real clinical data.

## Methods

### Model introduction

Machine learning model is an algorithm that learns and makes predictions or decisions by analyzing data, it analyzes and learns data to identify patterns and then uses the learned knowledge to make predictions or decisions on new data. There are many kinds of machine learning models, and the common ones include: linear regression, logistic regression, support vector, decision tree, neural network, and so on. The stacking ensemble model adopted in this paper is an integrated learning method, which improves the overall prediction performance by combining the prediction results of multiple different machine learning models. Its core concept is that various models excel in distinct aspects of data, by judiciously integrating these models, we can complement each other's strengths, mitigate the risk of overfitting in individual models, and achieve superior predictive performance to that of a single model, by offering more precise risk predictions, can assist doctors in making better clinical decisions and initiating early interventions for high-risk patients, potentially leading to a reduction in mortality rates. Simultaneously, stacking ensemble model exhibit flexibility and scalability, its structure allows researchers to add or replace base models or metas-models as needed, providing flexibility in model adjustment and optimization. In terms of statistics, the stacking ensemble model demonstrated its statistically significant advantages by comparing the performance with a single model, enhancing the confidence of the model results. In conclusion, the modeling approach employed in this study integrates various machine learning techniques, which not only enhances the accuracy and flexibility of predictions but also offers powerful data support for clinical decision-making.

### Data collection and data preprocessing

The clinical data used in this research comes from the People's Hospital of Liaoning Province (Shenyang, China), which is a prominent and comprehensive third class hospital in China, and has been rated as a professional demonstration center of the American Heart Association and a clinical research institution of cardiovascular drugs in China. The data used in this paper were from inpatients diagnosed with NSTEMI during the 5-year period between January 1 2017, and December 31, 2022. There were about 800 cases of NSETMI each year. Each patient's data was collected from the electronic information system of the hospital, including the electronic medical record system, hospital information system, laboratory information

system, ultrasound system and electrocardiogram system. Fig 1 shows the patient selection process of our study. We finally included 3061 NSTEMI patients, with 2911 in the surviving group and 150 in the death group. As shown in Table 1, a total of 73 features related to hospitalization were extracted. This study was retrospective and ultimately approved by the Ethics Committee of the People's Hospital of Liaoning Province (Ethics number: 2023-K021), and the dates when data were accessed for research purposes was February 10, 2023. Before analysis, the patient's data were data anonymized and deidentified.

## Overview of the data process framework

The framework structure of data processing consists of three parts, as shown in Fig 2. Firstly, the collected data are preprocessed and resampled using the synthetic minority over-sampling technique (SMOTE). Secondly, the feature selection is performed using the recursive elimination method (RFE) with the Area Under Curve (AUC) of the model as the criterion. Finally, a Stacking model based on multiple models is established to dynamically predict.

Data preprocessing includes the handling of missing data, balancing datasets, dataset partitioning, and standardization. The details are as follows:

1. Table 1 shows the missing feature variables. It can be seen that there are 5 features with a missing rate of 0, including Aobottom, AV, LA, RV, and LVPW. There are six features with a deletion rate of more than 30%, including Na, K, CL, Ca, Hypertension, and Diabetes. The number of remaining features with missing rates not exceeding 30% is 62. Features with a missing rate of more than 30% will have a significant impact on the subsequent analysis, so they are removed. For the features with missing values less than 30%, this paper uses multiple imputation (MI) to fill in. During the filling process, it does not use the predicted values as missing values, but instead uses the predicted values to find adjacent predicted values, maps them back to the original data, and uses the original data statistics to fill in. This method simulates the distribution trend of the missing data, so that the filled data can better maintain the original distribution of the data and the relationship between variables.

2. Since the number of NSTEMI surviving group included in the paper is 2911, and the number of death group is 150, the in-hospital mortality rate of NSTEMI is 4.9%. It can be seen

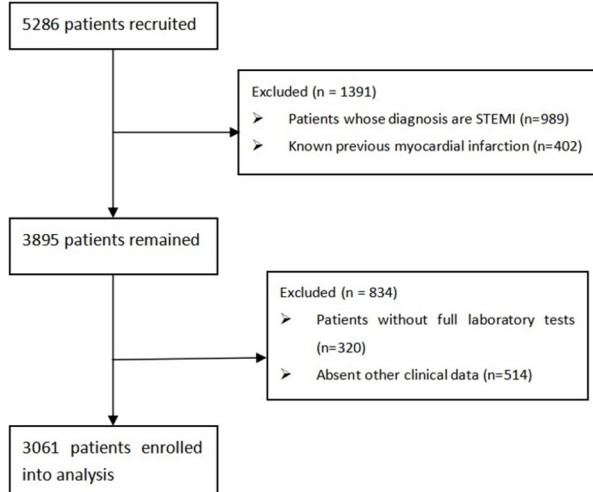

**Fig 1. Flow diagram of the patient selection process.**

**Table 1. Variable missing rate.**

| Categories | Missing rate | Categories | Missing rate | Categories | Missing rate | Categories | Missing rate |
|---|---|---|---|---|---|---|---|
| SEX_CODE | 21.1% | AGE | 2.6% | CRP | 2.6% | CREA | 4.4% |
| TNT_HS | 4.4% | cTNI | 7.3% | CK_MB | 7.2% | DD | 3.4% |
| NT_BNP | 3.4% | LDL C | 4.4% | HDL_C | 3.4% | NtoH | 3.4% |
| MtoH | 3.4% | NtoL | 3.4% | MtoL | 3.4% | CHOL | 3.4% |
| WBC | 3.4% | NEUTRATIO | 3.4% | LYMPHRATIO | 3.4% | MONORATIO | 3.4% |
| NEUNUM | 3.4% | LYMPHNUM | 3.4% | MONONUM | 3.4% | RBC | 3.4% |
| HGB | 3.4% | HCT | 11.3% | MCV | 11.3% | MCH | 11.3% |
| MCHC | 11.3% | Na | 34.4% | K | 31.9% | CL | 35% |
| Ca | 41% | RDW_CV | 11.3% | RDW_SD | 11.3% | Hypertension | 30.9% |
| Diabetes | 30.9% | ALT | 2.6% | AST | 4.4% | ALB | 5.2% |
| TBIL | 8.4% | DBIL | 1.4% | IBIL | 0.2% | BUN | 0.1% |
| TRIG | 0.2% | ZDB | 0.2% | UA | 0.1% | $CO^2$ | 0.1% |
| RVOT | 0.1% | AOtop | 0.1% | AOmid | 4.8% | AObottom | 0 |
| AV | 0 | LA | 0 | RV | 0 | IVS | 6.0% |
| LV | 6.1% | LVPW | 0 | EDV | 0.7% | ESV | 0.7% |
| EF | 1.6% | SV | 2.5% | EJBKEF | 2.6% | EJBKAF | 0.1% |
| EA | 0.04% | Smoke | 0.3% | Drink | 17.5% | HEIGHT | 17.5% |
| WEIGHT | 4.4% | BMI | 4.6% | Syspressure | 0.2% | Diapresure | 0.07% |
| HEARTRATE | 0.5% | | | | | | |

that there is a severe imbalance between positive and negative samples in the dataset. Using imbalanced data to train the learner will divide all samples into categories with large sample sizes, and no matter how high the accuracy of the of dead patients and survivors balanced, and the total number of balanced data sets is 5822.

3. The balanced data set is divided into training set and test set in a ratio of 8:2, in which the number of training sets is 4657, and the number of test sets is 1165. The number of positive samples in the training set is 2331 and the number of negative samples is 2326. The number of positive samples in the test set is 580 and the number of negative samples is 585. From the above data, it can be seen that the proportion of positive and negative samples in the training set and the test set is relatively balanced.

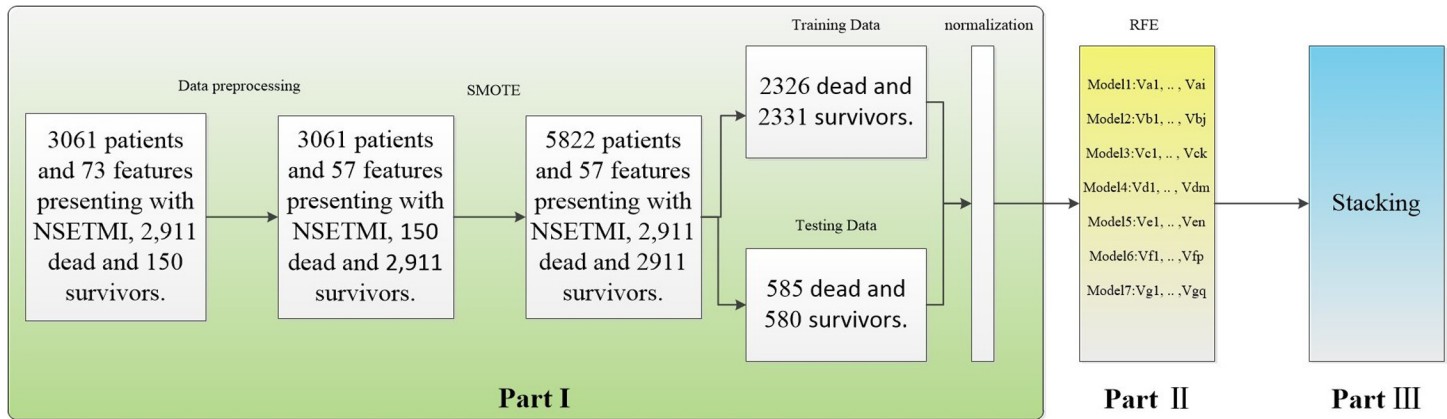

**Fig 2. Structural framework of the method.**

4. Standardization is to eliminate numerical differences between variables by normalizing all variables to a state with mean 0 and variance 1, as shown in Eq (1). x is the input, mean is the average and σ is the standard deviation, and $x^*$ is the normalized output.

$$x^* = \frac{x - mean}{\sigma} \tag{1}$$

## Feature selection

Feature selection is a process used to filter feature subsets that has an important impact on the prediction results. The recursive feature elimination (RFE) [20] is used as the method of feature extraction in this paper. This method is an important method in the nonlinear classifier of the package method, which can select high-quality subsets, but the speed is relatively slow because it integrates the feature selection process with the training process. Although this process is time- consuming, it is only completed once in the preprocessing, and no additional time is consumed in the subsequent prediction. This paper uses RFECV in scikit-learn in Python to implement the RFE method, and then uses cross-validation to select the optimal number of features after evaluating the importance of features. The specific steps are as follows:

1. Select all features as the initial feature set.

2. Use the current feature set to model and calculate the importance of each feature.

3. Update the dataset by removing the least important feature (or features).

4. Skip to step 2 until all important ratings are completed.

5. According to the feature importance determined in step 4, select different number of features in turn.

6. Perform cross-validation on the selected feature set.

7. Determine the number of features with the highest average score and complete feature selection.

## Model building

Since the RFE method is used for feature selection in this paper, which requires the model to have the attribute of feature importance, the model with feature importance as the preselection model, These include Logistic Regression (LR), Decision Tree (DT), Support Vector Machine (SVM), Random Forest (RF), Adaboost (ADB), Extra Trees (ET), Gradient Boosting Decision Tree (GBDT), and XGBOOST (XGB) [21]. And the model parameters are optimized by means of five-fold cross test, and the parameters of each model are shown in Table 2.

Firstly, the stacking ensemble model is used as the training model in this paper, which consists of a two-layer structure, the first layer is the base model, and the second layer is the meta-model. For the base model, 7 learners are selected and predicted by 5-fold cross detection stacking, while for the meta-model, XGB is used in this paper. The training set generated by the first layer is input to the second layer for the final prediction. The integrated model framework based on Stacking proposed in this paper is shown in Fig 3. Model[1] to Model[7] represent seven base models, $V_1$ to $V_n$ represent the features of the data, where $V_{a1}$ to $V_{ai}$ represent the

**Table 2. Parameters of the seven models.**

| Models | Parameter |
|--------|-----------|
| LR | C = 0.1 |
| DT | max depth = 5, min_sample_split = 10, random_state = 1 |
| SVM | C = 0.01, kernel = 'linear', probability = True |
| RF | max depth = 5, n_estimators = 80, random state = 1 |
| GBDT | max depth = 3, n_estimators = 200, learning rate = 0.01, random state = 1 |
| ET | bootstrap = False, max depth = 6, n_estimators = 200, random_state = 1 |
| ADB | N_estimators = 200, learning rate = 0.1, random = 1 |
| XGB | learnrate = 0.01, max depth = 4, n_estimators = 190, random_ state = 1 |

feature subset of Model[1], $V_{b1}$ to $V_{bj}$ represent the feature subset of Model[2], and so on. The detailed steps of the final Stacking ensemble model are described as follows.

In the first layer, for each base model (Model[1] to Model[7]), the corresponding feature subset is taken as input and the prediction of the base classifier is generated by a 5-fold cross test. As shown in Part B of Fig 3, we split the training set into five folds for cross-validation. In each iteration, 4 folds are used to train the classifier and the remaining 1 fold is used for prediction.

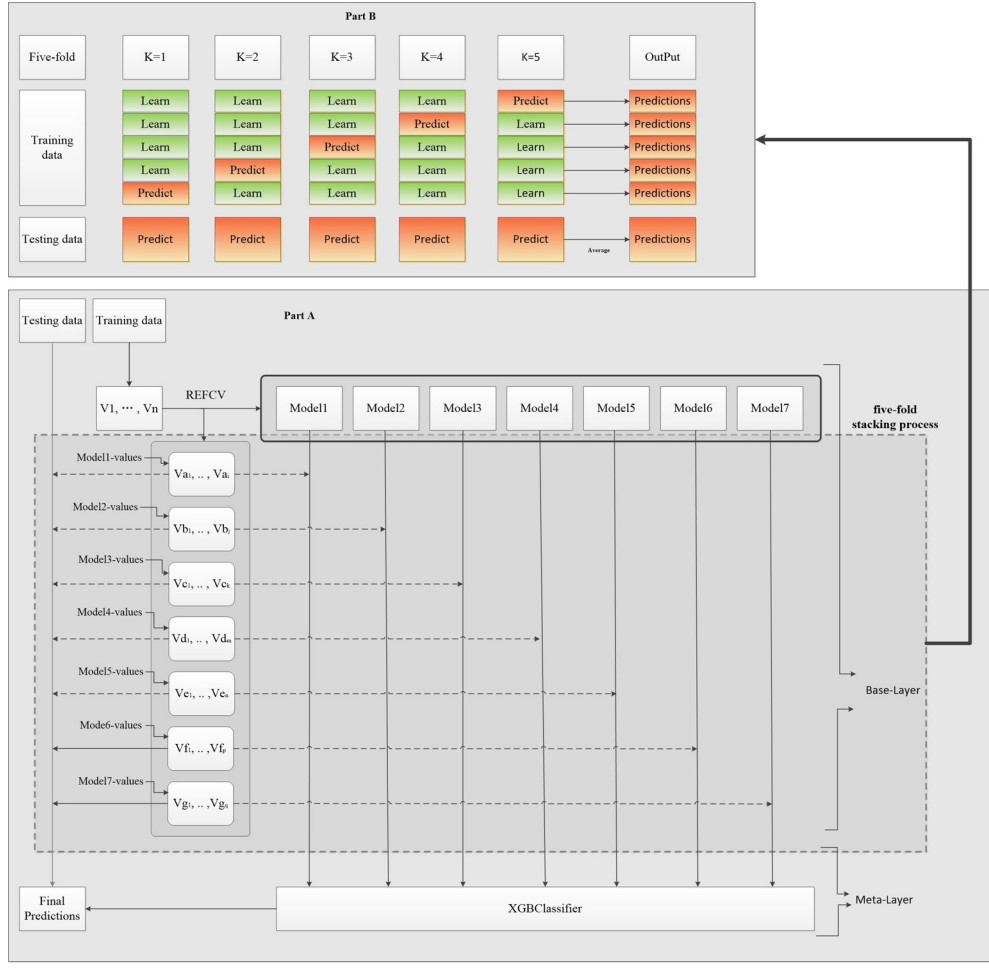

**Fig 3. Framework diagram of the proposed Stacking-based ensemble learning model.**

Meanwhile, in each iteration, the test set is predicted by the trained classifier. After five iterations, the prediction results for the training set can be obtained. The average of the prediction results of the classifier in the test set is used as the prediction result of the base model.

In the second layer, due to the best performance of XGB compared to other models, XGB is chosen as the meta-model to train the prediction results of the training set generated by the first layer, and the final prediction is made using the mean of the first layer's base model test set.

1. Evaluation metrics: In this paper, Area Under Curve (AUC) is taken as the main evaluation index. In order to more comprehensively evaluate the method proposed in this paper, the confusion matrix is further used as an evaluation index, which includes Accuracy, Precision, Recall and F1. As shown in Eqs (2)-(5), where TP = true positive, FP = false positive, TN = true negative and FN = false negative.

$$\text{Accuracy} = \frac{\text{TP} + \text{TN}}{\text{TP} + \text{FN} + \text{FP} + \text{TN}} \tag{2}$$

$$\text{Recall} = \frac{\text{TP}}{\text{TP} + \text{FN}} \tag{3}$$

$$\text{Precision} = \frac{\text{TP}}{\text{TP} + \text{FP}} \tag{4}$$

$$\text{F1} = \frac{2*\text{Precision}*\text{Rrcall}}{\text{Precision} + \text{Recall}} \tag{5}$$

## Statistical analysis

The experimental environment in this paper is implemented in Python 3.8.13. The packages of scikit-learn 1.1.1 and imblearn 0.9.1 are used for oversampling, feature selection, and ML algorithms. All analyses are performed on a computer running an AMD R7 3.20GHz processor with Windows 10 operating system and 16 gigabytes of memory. All categorical data are presented as percentages, all normally distributed continuous data are presented as mean (standard deviation), and all non-normally distributed continuous data are presented as median (IQR). T-test, non-parametric test and chi-square test are used for comparison between groups. And $P < 0.05$ is considered statistically significant.

## Results

### Baseline characteristics

A total of 3061 hospitalized patients diagnosed with NSTEMI are enrolled in this paper, and the in-hospital mortality rate is 4.9% (2911 in the survival group and 150 in the death group). The baseline characteristics of the eligible population are shown in Table 3. The age of patients in the death group (79 years, IQR 71–83 years) is greater than that in the survival group (67 years, IQR 72–83 years). CRP (median survival 3.83, IQR 0.56–16.21, median death 34.87, IQR 13.79–63.67), CREA (median survival 76.5, IQR 63.6–99.3, median death 147.67, IQR 105.47–130.44), TNT_HS(the median of the survival group is 262.7, IQR is 87.02–833.8, the median of the death group is 828.99, IQR is 296.54–2352.53, and $P$-value < 0.05) cTNI (the median of the survival group is 1220, IQR is 273–4780, the median of the death group is 2899.34, IQR is 477.85–19474.03, and $P$-value < 0.05), NT_BNP (the median of the survival group is 1095,

**Table 3. Clinical variables in the death and survivor groups (N = 3061).**

| Features | NSTEMI | | |
|---|---|---|---|
| | Survivor group | Death group | |
| * SEX CODE (men/women) | 1937 (66.5%)/974(33.5%) | 72(48%)/78(52%) | <0.01 |
| *AGE, median (IQR) | 67(59–77) | 79(72–83) | <0.01 |
| *CR, median (IQR) | 3.83(0.56–16.21) | 34.87(13.79–63.67) | <0.01 |
| *CRE, median (IQR) | 76.5(63.6–99.3) | 147.67(105.47–130.44) | <0.01 |
| *TNT_H, median (IQR) | 262.7(87.02–833.8) | 828.99(296.54–2352.53) | <0.01 |
| *cTNI, median (IQR) | 1220(273–4780) | 2899.34(477.85–19474.03) | <0.01 |
| *CK_MB, median (IQR) | 3.41(1.62–9.05) | 6.72(2.90–18.19) | <0.01 |
| *DD, median (IQR) | 0.39(0.21–0.94) | 2.12(0.98–5.61) | <0.01 |
| *NT_BNP, median (IQR) | 1095(338–3932) | 16362.13(6059.89–33604.05) | <0.01 |
| *LDL_C, median (IQR) | 2.70(2.11–3.31) | 2.4(1.88–3.18) | 0.011 |
| HDL_C, median (IQR) | 0.96(0.80–1.14) | 0.93(0.77–1.14) | 0.201 |
| *NtoH, median (IQR) | 5.43(3.92–7.35) | 10.16(7.39–13.41) | <0.01 |
| *MtoH, median (IQR) | 0.56(0.41–0.78) | 0.73(0.55–0.96) | <0.01 |
| *NtoL, median (IQR) | 3.26(2.315–4.88) | 9.65(6.20–16.53) | <0.01 |
| *MtoL, median (IQR) | 0.35(0.25–0.49) | 0.72(0.46–1.17) | <0.01 |
| *CHOL, median (IQR) | 4.32(3.61–5.06) | 4.07(3.37–5.03) | 0.035 |
| *WBC, median (IQR) | 7.61(6.30–9.32) | 10.84(8.45–13.94) | <0.01 |
| *NEUTRATIO, median (IQR) | 68.9(62.2–75.65) | 82.34(75.43–87.21) | <0.01 |
| *LYMPHRATIO, median (IQR) | 21.3(15.6–26.9) | 9.99(6.50–115.49) | <0.01 |
| *MONORATIO, median(IQR) | 7.1(5.8–8.7) | 6.15(4.59–7.87) | <0.01 |
| *NEUNUM, median (IQR) | 5.1(4–6.73) | 8.48(6.17–11.48) | <0.01 |
| *LYMPHNUM, median (IQR) | 1.57(1.17–2.03) | 0.98(0.69–1.41) | <0.01 |
| *MONONUM, median(IQR) | 0.53(0.4–0.7) | 0.64(0.48–0.83) | <0.01 |
| *RBC, median (IQR) | 4.32(3.85–4.73) | 3.67(3.16–4.09) | <0.01 |
| *HGB, median (IQR) | 133(118–1146) | 109(93–124) | <0.01 |
| *HCT, median (IQR) | 39.4(35.1–42.9) | 33.24(28.78–38.58) | <0.01 |
| *MCV, median (IQR) | 91(88–94) | 92.7(88.88–96.10) | <0.01 |
| MCH, median (IQR) | 30.9(29.6–32) | 30.45(29.36–31.57) | 0.055 |
| *MCHC, median (IQR) | 337(330–344) | 328(322–335) | <0.01 |
| *RDW_CV, median (IQR) | 17.7(12.2–13.4) | 14.11(13.25–15.29) | <0.01 |
| *RDW_SD, median (IQR) | 42.3(40.1–45.1) | 46.99(44.36–50.63) | <0.01 |
| *ALT, median (IQR) | 22.1(15–35.3) | 31.79(19.3–65.79) | <0.01 |
| *AST, median (IQR) | 30.1(20.9–53.4) | 61.95(29.99–144.57) | <0.01 |
| *ALB, median (IQR) | 37.9(35.1–40.4) | 33.79(31.07–36.06) | <0.01 |
| TBIL, median (IQR) | 13.8(10.6–18.4) | 14.21(10.84–18.15) | 0.496 |
| *DBIL, median (IQR) | 2.7(2–3.8) | 3.31(2.46–5.34) | 0.001 |
| IBIL, median (IQR) | 11(8.4–14.7) | 10.6(8.0–13.9) | 0.137 |
| *BUN, median (IQR) | 6.05(4.9–8.355) | 13.99(9.33–20.98) | <0.01 |
| *TRIG, median (IQR) | 1.36(0.98–1.94) | 1.24(0.88–1.67) | 0.011 |
| ZDB, median (IQR) | 157 (78.15–307.45) | 157.6(86.08–359.63) | 0.288 |
| *UA, median (IQR) | 360(292–445) | 448.92(361–594.57) | <0.01 |
| *$CO^2$, median (IQR) | 36.5(24.3–28.7) | 25.59(22.79–27.90) | <0.01 |
| RVOT, median (IQR) | 30(30–30) | 30(29–30) | 0.964 |
| AOtop, median (IQR) | 22(22–23) | 22(21–23) | 0.389 |
| AOmid, median (IQR) | 34(32–36) | 33(32–36) | 0.279 |
| AObottom, median (IQR) | 30(30–32) | 30(29.25–32) | 0.284 |

*(Continued)*

**Table 3.** (Continued)

| Features | NSTEMI | | |
|---|---|---|---|
| | Survivor group | Death group | |
| AV, median (IQR) | 19(19–19) | 19(19–19) | 0.151 |
| *LA, median (IQR) | 40(38–44) | 42(39–46) | <0.01 |
| *RV, median (IQR) | 18(18–18) | 18(18–18) | <0.01 |
| *IVS, median (IQR) | 11(10–12) | 10(10–11) | <0.01 |
| *LV, median (IQR) | 49(46–53) | 52(49–56) | <0.01 |
| *LVPW, median (IQR) | 10(10–11) | 10(9–10) | <0.01 |
| *EDV, median (IQR) | 102(88–132) | 126(105–154) | <0.01 |
| *ESV, median(IQR) | 54(42–79) | 83.54(64.93–107.23) | <0.01 |
| *EF, median (IQR) | 0.45(0.39–0.55) | 0.34(0.29–0.40) | <0.01 |
| * SV, median (IQR) | 48(41–56) | 43.14(36.90–49.68) | <0.01 |
| *EJBKEF, median (IQR) | 70(60–90) | 86(67–106) | <0.01 |
| *EJBKAF, median (IQR) | 90(70–100) | 80.80(67.99–97.59) | <0.01 |
| *EA, (0/0.8) | 2537(87.2%)/288(9.9%) | 100(66.7%)/40(26.7%) | <0.01 |
| *Smoke (y/n) | 1318(45.3%)/ 1593(54.7%) | 51(34%)/99(66%) | <0.01 |
| *Drink (y/n) | 778(26.7%) / 2133(73.3%) | 18(12%)/132(88%) | <0.01 |
| *HEIGHT, median (IQR) | 1.69(1.60–1.73) | 1.64(1.59–1.69) | <0.01 |
| *WEIGHT, median (IQR) | 70(60–78) | 62.21(55.15–68.16) | <0.01 |
| *BMI, median (IQR) | 24.77(22.60–27.05) | 22.91(21.13–24.64) | <0.01 |
| * Syspressure, median (IQR) | 139(124–157) | 127(110–141) | <0.01 |
| *Diapresure, median (IQR) | 79(70–89) | 70(62–78) | <0.01 |
| *HEARTRATE, median (IQR) | 79(70–92) | 94(80–105) | <0.01 |

IQR is 338–3932, the median of the death group is 16362.13, IQR:6059.89–33604.05) and other indicators can be seen that the value of the death group is much higher than that of the survival group, and the comparison between the groups of these indicators is all $P < 0.05$, indicating that these indicators are highly correlated with the in-hospital death of NSTEMI patients. Among them, the P values of 10 features such as HDL_C, MCH, TBIL, IBIL, ZDB, RVOT, AOtop, AOmid, AObottom and AV are all $> 0.05$ and has no statistical significance, so these features are excluded. The number of features finally included in the subsequent model training is 57.

## Oversampling result

Before the SMOTE oversampling, there are 3061 NSTEMI patient data, of which 2911 are in the survivor group and 150 in the death group. AUC and F1 are used to compare the performance of the eight models before and after SMOTE. The results are shown in Table 4. It can be seen that the AUC and F1 indicators of XGB are the highest among the eight models. Therefore, XGB is chosen as the meta-model for Stacking in this paper.

## Results of feature selection

The dataset has 57 features after the above processing. After feature selection using machine learning methods, the number of features required in the model can be further reduced to a certain extent. The results of RFE feature selection using seven different methods are shown in Table 5. It can be seen from the table that the number of features for LR, DT, SVM, RF, GBDT, ET and ADB are 42, 25, 38, 49, 39, 49 and 39 respectively. Each model has removed a large

**Table 4. Comparison of indicators before and after SMOTE.**

| models | AUC | | F1 | |
|---|---|---|---|---|
| | Before | After | Before | After |
| LR | 0.895 | 0.933 | 0.324 | 0.866 |
| DT | 0.697 | 0.945 | 0.364 | 0.926 |
| SVM | 0.898 | 0.942 | 0.082 | 0.886 |
| RF | 0.894 | 0.948 | 0.235 | 0.889 |
| GBDT | 0.875 | 0.920 | 0.286 | 0.855 |
| ET | 0.906 | 0.938 | 0.065 | 0.863 |
| ADB | 0.868 | 0.949 | 0.410 | 0.880 |
| XGB | 0.898 | 0.964 | 0.439 | 0.941 |

number of redundant features, especially for DT, SVM, GBDT, and ADB, which are 25, 38, 39 and 39 respectively, greatly reducing model training time. Taking accuracy and times as evaluation criteria, the performance of seven candidate models before and after RFE is compared, as shown in Table 6. It can be seen that among the seven models, the performance of the model after RFE feature selection is mostly better than that before RFE.

## Comparison of the performance of the different predictive models

In order to evaluate the predictive performance of the Stacking-based model proposed in this paper, we compared it with these seven models. The comparison results of the performance indicators of different models are shown in Table 7. From the table, it can be seen that the Stacking model proposed in this paper has a good performance in all evaluation indicators. The AUC, Accuracy, Precision, Recall and F1 are 0.987, 0.942, 0.959, 0.945 and 0.941, respectively. Fig 4 shows the comparison of the ROC curves of the model. For the area AUC below the ROC curve, compared with the best model ADB, the Stacking model improves the AUC

**Table 5. The number of features of the model after RFE feature selection.**

| Model | Nums | Features |
|---|---|---|
| LR | 42 | SEX_CODE, AGE, CRP, TNT_HS, cTNI, CK_MB, NT_BNP, LDL_C, NtoH, NtoL, MtoL, CHOL, WBC, NEUTRATIO, LYMPHRATIO, MONORATIO, LYMPHNUM, MONON, UM, MCV, MCHC, ALB, DBIL, BUN, TRIG, UA, $CO^2$, LA, RV, IVS, LV, LVPW, EDV, EF, SV, EJBKAF, Smoke, Drink, HEIGHT, WEIGHT, BMI, Syspressure, Diapresure |
| DT | 25 | CRP, TNT_HS, CK_MB, DD, NT_BNP, MtoH, NtoL, WBC, LYMPHRATIO, MONORATIO, RBC, MCHC, RDW_SD, AST, ALB, DBIL, BUN, TRIG, LVPW, ESV, SV, EA, Drink, HEIGHT, BMI |
| SVM | 38 | SEX_CODE, AGE, CRP, TNT_HS, CK-MB, NT_BNP, LDL_C, NtoH, NtoL, MtoL, CHOL, WBC, NEUTRATIO, LYMPHRATIO, MONORATIO, LYMPHNUM, MONONUM, HGB, MCV, MCHC, ALB, BUN, TRIG, UA, LA, RV, IVS, LVPW, EDV, EF, EJBKAF, Smoke, Drink, HEIGHT, WEIGHT, BMI, Syspressure, Diapresure |
| RF | 49 | SEX_CODE, AGE, CRP, CREA, TNT_HS, cTNI, CK_MB, DD, NT_BNP, LDL_C, NtoH, MtoH, NtoL, MtoL, WBCNEUTRATIO, LYMPHRATIO, NEUNUM, LYMPHNUM, MONONUM, RBC, HGB, HCT, MCHC, RDW_CV, RDW_SD, ALT, AST, ALB, DBIL, BUN, TRIG, IVS, LVPW, EDV, ESV, EF, SV, EJBKEF, EJBKAF, EA, Smoke, Drink, HEIGHT, WEIGHT, BMI, Syspressure, Diapresure, HEARTRATE |
| GBDT | 39 | SEX_CODE, CRP, CREA, TNT_HS, CK_MB, NT_BNP, LDL_C, NtoH, NtoL, MtoL, WBC, LYMPHRATIO, MONORATIO, NEUNUM, RBC, HGB, MCV, RDW_CV, RDW_SD, ALT, AST, ALB, DBIL, BUN, TRIG, IVS, LVPW, ESV, EF, SV, EJBKEF, EJBKAF, EA, Drink, HEIGHT, WEIGHT, Syspressure, Diapresure |
| ET | 49 | SEX_CODE, AGE, CRP, CREA, TNT_HS, CK_MB, CK_MB, NT_BNP, NtoH, MtoH, NtoL, MtoL, WBC, NEUTRATIO, LYMPHRATIO, MONORATIO, NEUNUM, LYMPHNUM, MONONUM, RBC, HGB, HCT, MCHC, RDW_CV, RDW_SD, ALT, AST, ALB, BUN, UA, IVS, LV, LVPW, EDV, ESV, EF, SV, EJBKEF, EJBKAF, EA, Smoke, Drink, HEIGHT, WEIGHT, BMI, Syspressure, Diapresure, HEARTRATE |
| ADB | 39 | SEX_CODE, CRP, CREA, TNT_HS, CK_MB, CK_MB, DD, NT_BNP, LDL_C, NtoH, MtoH, NtoL, MtoL, CHOL, WBC, MONORATIO, MCHC, RDW_CV, RDW_SD, ALT, AST, ALB, DBIL, BUN, TRIG, UA, IVS, LVPW, ESV, EF, SV, EJBKEF, EJBKAF, EA, Smoke, Drink, HEIGHT, BMI, Syspressure, Diapresure |

Table 6. Comparison of results before and after RFE.

| models | Accuracy | | times | |
|---|---|---|---|---|
| | Before | After | Before | After |
| LR | 0.847 | 0.870 | 0.027s | 0.019s |
| DT | 0.881 | 0.926 | 0.249s | 0.099s |
| SVM | 0.879 | 0.884 | 1.466s | 1.224s |
| RF | 0.858 | 0.888 | 0.545s | 0.573s |
| GBDT | 0.841 | 0.858 | 8.020s | 5.910s |
| ET | 0.850 | 0.860 | 0.433s | 0.422s |
| ADB | 0.869 | 0.877 | 3.511s | 2.557s |

index by 4.0%, the Accuracy by 7.4%. Precision improved by 8.4%, Recall improved by 7.0%, and F1 improved by 6.9%, respectively.

## Discussion

This paper proposes a prediction model based on the Stacking model to predict the in-hospital mortality of NSTEMI patients. The model uses 57 features, all of which can be obtained from the hospital's electronic medical record system, hospital information system, laboratory information system, ultrasound system and electrocardiogram system. Due to the more detailed data obtained, compared with the traditional risk assessment method, this paper incorporates more features to predict the in-hospital mortality of NSTEMI patients. Compared with a single ML algorithm, there is a significant improvement in performance, which can make predictions more accurately and timely remind doctors to provide medical assistance to patients.

Although NSTEMI presents a high risk for death and cardiac ischemic events, the degree of clinical attention is often not as high as ST-segment elevation myocardial infarction (STEMI). In-hospital mortality risk analysis is very meaningful for NSTEMI patients, as the prognosis of NSTEMI is highly uncertain. Therefore, early detection and timely diagnosis and treatment have important clinical significance. At present, many models have been proposed to predict the risk of death in acute coronary syndrome (ACS), and GRACE score and TIMI score are the most commonly used tools in clinical practice [22]. However, both scores have limitations. The TIMI score is based on clinical trial data and has strict inclusion and exclusion criteria, therefore its performance is limited. The patient population with the GRACE score mainly comes from the United States, Europe, and Australia, not from the Asian population. Furthermore, the GRACE score was obtained from a GRACE-enrolled cohort of ACS patients more than twenty years ago. As time goes on, there have been significant developments and changes in the management of patient records and AMI [23]. With the rise of artificial intelligence,

Table 7. Comparison of performance metrics of different models.

| models | AUC | Accuracy | Recall | Precision | F1 |
|---|---|---|---|---|---|
| LR | 0.934 | 0.870 | 0.834 | 0.900 | 0.867 |
| DT | 0.946 | 0.926 | 0.918 | 0.934 | 0.926 |
| SVM | 0.942 | 0.884 | 0.899 | 0.874 | 0.886 |
| RF | 0.948 | 0.888 | 0.899 | 0.880 | 0.889 |
| GBDT | 0.920 | 0.858 | 0.834 | 0.878 | 0.855 |
| ET | 0.938 | 0.860 | 0.877 | 0.849 | 0.863 |
| ADB | 0.949 | 0.877 | 0.896 | 0.865 | 0.880 |
| Our method | 0.987 | 0.942 | 0.959 | 0. 945 | 0.941 |

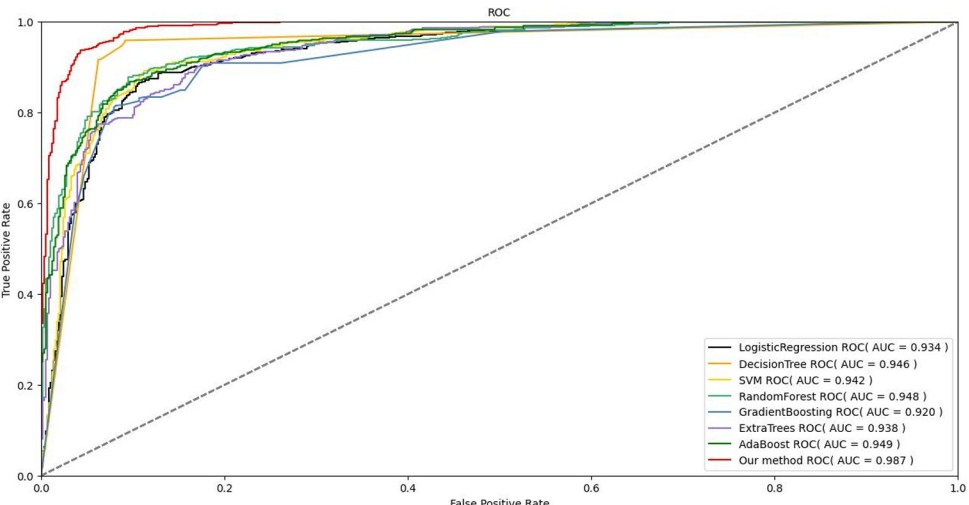

**Fig 4. ROC curve of the model.**

there are more and more researches on its application in the medical field [24]. For example, heart disease can be predicted by combining machine learning and deep learning [25]. Training machine learning to improve the predictive ability of COVID-19 [19], etc. At present, there are few studies on the prediction of in-hospital death risk of NSTEMI patients with the machine learning method.

In this study, we excluded 834 individuals due to incomplete laboratory or clinical data. The reason is that essential values typically refer to variables that are crucial to the research question. If these variables are missing, the data from these individuals may not support the research purpose and thus need to be addressed. Additionally, excluding these individuals helps to avoid bias, ensure data consistency, meet research hypotheses, improve analysis efficiency, and ensure the interpretability of the results. Therefore, we excluded individuals with missing essential data.

Some of the characteristics used in this paper are consistent with those used in previous models, including age, gender, and BMI. Compared with the previous studies, this model adds laboratory information, including troponin, creatinine, C-reactive protein and other variables associated with NSTEMI. As shown in Table 3, It can be seen that TNT-HS and cTNI are statistically significant in the death group and the survival group, and these two variables are closely related to the in-hospital death of NSTEMI patients. Similarly, CRP and CREA are significantly associated with in-hospital mortality in NSTEMI patients. In recent years, the relationship between Monocyte/High-density lipoprotein cholesterol ratio (MtoH), neutrophil/lymphocyte ratio (NtoL), Monocyte/lymphocyte ratio (MtoL) and neutrophil/High-density lipoprotein cholesterol ratio (NtoH) with AMI patients is being studied by experts and scholars [26]. These four indicators are also included in this paper, and the comparison between groups is $P < 0.05$, which is statistically significant. After RFE feature selection, it is found that LR model contains NtoH, NtoL and MtoL, DT model contains MtoH and NtoL, SVM model contains NtoH, NtoL and MtoL, RF model contains NtoH, MtoH, NtoL and MtoL, GBDT contains NtoH, NtoL and MtoL, and GBDT contains NTOH, NTOL and MTOL. ET contains NtoH, MtoH, NtoL, and MtoL, and ADB contains NtoH, MtoH, NtoL, and MtoL. The $P$ values of NtoH, MtoH, NtoL, and MtoL compared between the survival groups and the death groups are all $< 0.05$ in Table 3. Therefore, it can be proved that NtoH, MtoH, NtoL, and MtoL are important indicators in predicting in-hospital mortality.

The performance of ADB is the best among the seven candidate models in this paper, so it is inferred that ADB may have prominent performance in predicting the in-hospital mortality of patients. Compared with ADB, the AUC, Accuracy and F1 of the Stacking model proposed in this paper are increased by 4.0%, 7.4% and 6.9%, respectively. It is shown that the model proposed in this paper can further improve the overall prediction performance compared to the best individual models. The Stacking model uses the predictions of multiple base learners as features to train a new meta-learner. If the base classifier performs poorly, it may not achieve good prediction performance. On the other hand, the criteria for adjusting the parameters of each candidate model and selecting the basic classifier in this paper are based on the AUC index, rather than F1. AUC is used as an evaluation index for binary classification, and using this evaluation indicator as a standard training model can achieve the optimal classification results.

This paper presents a model for predicting in-hospital mortality of patients. The prediction results of this model can help doctors make judgments on patients' conditions faster and adjust the allocation of medical resources for better treatment of patients. This paper includes all available clinical data, such as demographic information, hospitalization information, physical examination information, ultrasound examination information, and laboratory testing and other detailed clinical data, which are not available in traditional scores such as GRACE score and TIMI score. Adding these clinical data and applying it to training prediction models makes the model in this paper more convincing.

There are still some limitations in this paper. First, since the feature selection method of RFE requires the attributes of feature importance, this paper does not include models without feature importance attributes (such as artificial neural networks, nonlinear kernel SVM), which limits the ability to compare with more different types of models. Second, due to the large time cost of RFE feature selection, it is necessary to further explore effective ways to save time in order to obtain the optimal feature subset of the model faster. Third, some long text information is not included in this paper (such as Killip classification, discharge summary, etc.), otherwise this paper can obtain the time change information of some indicators of patients during hospitalization, so as to further improve the accuracy of the model.

## Conclusion

This paper proposes an XGB-based Stacking ensemble model for predicting NSETMI in-hospital mortality risk based on clinical data. The results of comparison with the single ML model show that the proposed model has outstanding performance in the evaluation indicators of Accuracy, AUC, Precision, Recall, and F1, indicating that the Stacking ensemble model proposed in this paper can integrate the advantages of different individual models to achieve better predictive performance. In addition, the proposed Stacking model is developed with real clinical data, which makes the model more convincing.

An effective mortality risk prediction model can provide valuable insights for doctors to diagnose patients' conditions and provide early clinical intervention for high-risk patients to avoid patient death as much as possible. In future research, the proposed Stacking ensemble model can also be further optimized by integrating data from more hospitals.

## Supporting information

**S1 File.**
(DOC)

## Acknowledgments

The authors would like to thank the patients, doctors and engineers who participated in this study.

## Author Contributions

**Conceptualization:** Li Wang, Feng li, Hongzeng Xu.

**Data curation:** Yu zhang, Feng li.

**Funding acquisition:** Hongzeng Xu.

**Investigation:** Li Wang.

**Methodology:** Li Wang, Yu zhang, Caiyun Li, Hongzeng Xu.

**Resources:** Caiyun Li.

**Supervision:** Feng li, Hongzeng Xu.

**Writing – original draft:** Li Wang, Feng li.

**Writing – review & editing:** Caiyun Li.

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
