## [Decision Letter · Decision Letter 0]

15 Jul 2024

PONE-D-24-07868Mortality Prediction of Inpatients with NSTEMI Based on Stacking ModelPLOS ONE

Dear Dr. hongzeng,

Thank you for submitting your manuscript to PLOS ONE. After careful consideration, we feel that it has merit but does not fully meet PLOS ONE’s publication criteria as it currently stands. Therefore, we invite you to submit a revised version of the manuscript that addresses the points raised during the review process.

We look forward to receiving your revised manuscript.

Kind regards,

Pasyodun Koralage Buddhika Mahesh

Academic Editor

PLOS ONE

2. In this instance it seems there may be acceptable restrictions in place that prevent the public sharing of your minimal data. However, in line with our goal of ensuring long-term data availability to all interested researchers, PLOS’ Data Policy states that authors cannot be the sole named individuals responsible for ensuring data access (http://journals.plos.org/plosone/s/data-availability#loc-acceptable-data-sharing-methods).

Reviewers' comments:

Reviewer's Responses to Questions

**Comments to the Author**

1. Is the manuscript technically sound, and do the data support the conclusions?

Reviewer #1: Partly

Reviewer #2: Yes

Reviewer #3: Partly

2. Has the statistical analysis been performed appropriately and rigorously? 

Reviewer #1: I Don't Know

Reviewer #2: Yes

Reviewer #3: Yes

3. Have the authors made all data underlying the findings in their manuscript fully available?

Reviewer #1: Yes

Reviewer #2: Yes

Reviewer #3: No

4. Is the manuscript presented in an intelligible fashion and written in standard English?

Reviewer #1: No

Reviewer #2: Yes

Reviewer #3: Yes

5. Review Comments to the Author

Reviewer #1: Since this paper is more towards advanced statistical methods on model building for mortality prediction among the NSTEMI patients, I feel that the authors shall explain about the statistical methods bit more so that any medical professionals with little background of statistical knowledge could grab the essence of this paper. Hence, it is always preferred to indicate the descriptives of the socio-demographic data of the groups (STEMI survivors and Deaths) and main variables according to the sex of the group members. It is worthwhile indicating how the gender sensitivity is alleviated in the prediction models when pooling the data (ex: Normalization procedure)

Unfortunately, I could not locate table 1 in the supplement given.

Reviewer #2: This study presents a novel stacking ensemble model to predict in-hospital mortality for patients with non-ST-segment elevation myocardial infarction. Utilising data from 3061 patients, the model combines various machine learning techniques to enhance prediction accuracy. The proposed model outperforms traditional methods, providing valuable insights for clinical decision-making and early interventions to reduce mortality rates.

Thank you for the opportunity to review this manuscript. I believe it is well-written and offers valuable insights. I have no further feedback and recommend it for publication.

Reviewer #3: The authors have conducted a mortality prediction exercise of inpatients with NSTEMI in People’s Hospital of Liaoning Province (Shenyang, China) based on Stacking Model. This model has shown higher performance in terms of AUC, Accuracy, Precision, Recall and F1 evaluation indicators. While congratulating the authors for this worthy study, following comments are given with the hope these will be beneficial for them.

1. Title should reflect the study setting. Suggested title is “Mortality Prediction of Inpatients with NSTEMI in premier hospital in Liaoning Province, China Based on Stacking Model”

2.In the methods section, give a brief account on the annual turnover of the patients with NSTEMI in this hospital, for the reader to get an understanding on any potential selection bias.

3. More clarity is needed on the participant selection process as shown in Figure 1. Do 5286 include all inpatients within those 5 years? However, 989 have been excluded for being not diagnosed as AMI. Please provide more details on this.

4. Another 834 have been excluded for the study for having incomplete laboratory or clinical data. Please discuss the implications of this in the discussion section.

5. It is mentioned that “In supplement, Table 1 shows the missing feature variables”. However this Table is not visible.

6. It is further mentioned that “ Features with a missing rate of more than 30% will have a significant impact on the subsequent analysis, so they are removed”. Discuss the implications of this step in the discussion section.

7. For the general readers to get a better understanding, it would be nicer to add a simple description of the performance parameters (i.e. AUC, Accuracy, Recall, Precision and F1).

6. PLOS authors have the option to publish the peer review history of their article (what does this mean?). If published, this will include your full peer review and any attached files.

Reviewer #1: No

Reviewer #2: **Yes: **Sameera Jayan Senanayake

Reviewer #3: **Yes: **I.O.K.Kumudumalee Nanayakkara

---

## [Author Response · Author response to Decision Letter 0]

15 Sep 2024

Replies to Comments of reviewers, and point-by-point responses:

We thank the reviewers for the time invested in critically reviewing our work and for their positive and highly valuable comments that have in our opinion significantly improved our paper. We have carefully studied the reviewers’ comments and thoughtful suggestions, responded to these suggestions point-by-point, and revised the manuscript accordingly. Please find below our responses to the specified points that were brought up by the reviewers, with a denotation of the changes made to the paper accordingly. With regard to the reviewers’ comments and suggestions, we’d like reply as follows:

Replies to reviewer 1

1.Since this paper is more towards advanced statistical methods on model building for mortality prediction among the NSTEMI patients, I feel that the authors shall explain about the statistical methods bit more so that any medical professionals with little background of statistical knowledge could grab the essence of this paper. Hence, it is always preferred to indicate the descriptives of the socio-demographic data of the groups (STEMI survivors and Deaths) and main variables according to the sex of the group members. It is worthwhile indicating how the gender sensitivity is alleviated in the prediction models when pooling the data (ex: Normalization procedure)

Response: Thank you for your suggestion. We have added a brief introduction to machine models and stacked ensemble models, as well as their advantages, at the beginning of the methods section, hoping that the public can gain a basic understanding of the statistical methods used in this paper.

 "Machine learning model is an algorithm that learns and makes predictions or decisions by analyzing data, it analyzes and learns data to identify patterns and then uses the learned knowledge to make predictions or decisions on new data. There are many kinds of machine learning models, and the common ones include: linear regression, logistic regression, support vector, decision tree, neural network, and so on. The stacking ensemble model adopted in this paper is an integrated learning method, which improves the overall prediction performance by combining the prediction results of multiple different machine learning models. Its core concept is that various models excel in distinct aspects of data, by judiciously integrating these models, we can complement each other's strengths, mitigate the risk of overfitting in individual models, and achieve superior predictive performance to that of a single model, by offering more precise risk predictions, can assist doctors in making better clinical decisions and initiating early interventions for high-risk patients, potentially leading to a reduction in mortality rates. Simultaneously, stacking ensemble model exhibit flexibility and scalability, its structure allows researchers to add or replace base models or metas-models as needed, providing flexibility in model adjustment and optimization. In terms of statistics, the stacking ensemble model demonstrated its statistically significant advantages by comparing the performance with a single model, enhancing the confidence of the model results. In conclusion, the modeling approach employed in this study integrates various machine learning techniques, which not only enhances the accuracy and flexibility of predictions but also offers powerful data support for clinical decision-making."

2.Unfortunately, I could not locate table 1 in the supplement given.

Response: Thank you for your valuable and thoughtful suggestion,we have added Table 1 to the attached schedule.

Reviewer #2: This study presents a novel stacking ensemble model to predict in-hospital mortality for patients with non-ST-segment elevation myocardial infarction. Utilising data from 3061 patients, the model combines various machine learning techniques to enhance prediction accuracy. The proposed model outperforms traditional methods, providing valuable insights for clinical decision-making and early interventions to reduce mortality rates. Thank you for the opportunity to review this manuscript. I believe it is well-written and offers valuable insights. I have no further feedback and recommend it for publication.

Response: Thank you very much for your positive evaluation of our research; it is a great encouragement for us.

Reviewer #3: The authors have conducted a mortality prediction exercise of inpatients with NSTEMI in People’s Hospital of Liaoning Province (Shenyang, China) based on Stacking Model. This model has shown higher performance in terms of AUC, Accuracy, Precision, Recall and F1 evaluation indicators. While congratulating the authors for this worthy study, following comments are given with the hope these will be beneficial for them.

1. Title should reflect the study setting. Suggested title is “Mortality Prediction of Inpatients with NSTEMI in premier hospital in Liaoning Province, China Based on Stacking Model”

Response: Thank you for your suggestion. We changed the title to "Mortality Prediction of Inpatients with NSTEMI in a premier hospital in China Based on Stacking Model" follow your advice.

2.In the methods section, give a brief account on the annual turnover of the patients with NSTEMI in this hospital, for the reader to get an understanding on any potential selection bias.

Response: Thank you for your suggestion. We have added the description of the patients. "There were about 800 cases of NSETMI each year". 

3. More clarity is needed on the participant selection process as shown in Figure 1. Do 5286 include all inpatients within those 5 years? However, 989 have been excluded for being not diagnosed as AMI. Please provide more details on this.

Response: Thank you for carefully and critically reviewing our manuscript. Our big data query system, but found that some inaccurate, so ruled out. We identified 5286 patients with myocardial infarction through the big data query system in the early stage, and then manually excluded 989 STEMI patients through clinical diagnosis. We modified the description in Figure 1.

4. Another 834 have been excluded for the study for having incomplete laboratory or clinical data. Please discuss the implications of this in the discussion section.

Response: Thank you for your question. Following your suggestion, we have discussed why we excluded the 834 individuals with missing laboratory and clinical data, and listed the advantages of doing so.

5. It is mentioned that “In supplement, Table 1 shows the missing feature variables”. However this Table is not visible.

Response: Thank you for your comments, and we have added Table 1 to the schedule according to your opinion.

6. It is further mentioned that “ Features with a missing rate of more than 30% will have a significant impact on the subsequent analysis, so they are removed”. Discuss the implications of this step in the discussion section.

Response: Thank you for your suggestions. Following your advice, we have added a discussion on why we chose to delete variables with more than 30% missing values, listed the advantages and disadvantages, and ultimately decided on this approach after weighing the pros and cons.

"In this study, we excluded 834 individuals due to incomplete laboratory or clinical data. The reason is that essential values typically refer to variables that are crucial to the research question. If these variables are missing, the data from these individuals may not support the research purpose and thus need to be addressed. Additionally, excluding these individuals helps to avoid bias, ensure data consistency, meet research hypotheses, improve analysis efficiency, and ensure the interpretability of the results. Therefore, we excluded individuals with missing essential data."

7. For the general readers to get a better understanding, it would be nicer to add a simple description of the performance parameters (i.e. AUC, Accuracy, Recall, Precision and F1).

 Response: Thank you for your comments, because there are many variables and methods studied, in order to increase readability, we put the indicators in Table 7.

---

## [Decision Letter · Decision Letter 1]

8 Oct 2024

Mortality Prediction of Inpatients with NSTEMI in a premier hospital in China Based on Stacking Model

PONE-D-24-07868R1

Dear Dr. hongzeng,

We’re pleased to inform you that your manuscript has been judged scientifically suitable for publication and will be formally accepted for publication once it meets all outstanding technical requirements.

Kind regards,

Pasyodun Koralage Buddhika Mahesh

Academic Editor

PLOS ONE

Additional Editor Comments (optional):

Reviewers' comments:

Reviewer's Responses to Questions

**Comments to the Author**

1. If the authors have adequately addressed your comments raised in a previous round of review and you feel that this manuscript is now acceptable for publication, you may indicate that here to bypass the “Comments to the Author” section, enter your conflict of interest statement in the “Confidential to Editor” section, and submit your "Accept" recommendation.

Reviewer #1: All comments have been addressed

Reviewer #3: All comments have been addressed

2. Is the manuscript technically sound, and do the data support the conclusions?

Reviewer #1: Yes

Reviewer #3: Yes

3. Has the statistical analysis been performed appropriately and rigorously? 

Reviewer #1: Yes

Reviewer #3: Yes

4. Have the authors made all data underlying the findings in their manuscript fully available?

Reviewer #1: Yes

Reviewer #3: Yes

5. Is the manuscript presented in an intelligible fashion and written in standard English?

Reviewer #1: Yes

Reviewer #3: Yes

6. Review Comments to the Author

Reviewer #1: All the comments therein have been attended adequately and let me congratulate the authors for this mater piece of research.

Reviewer #3: I would like to extend my congratulations to the authors for addressing all the comments and suggestions provided during the review process. The revisions made have enhanced the clarity, rigor, and overall quality of the research article titled "Mortality Prediction of Inpatients with NSTEMI in a Premier Hospital in China Based on Stacking Model."

7. PLOS authors have the option to publish the peer review history of their article (what does this mean?). If published, this will include your full peer review and any attached files.

Reviewer #1: No

Reviewer #3: **Yes: **I.O.K.K.Nanayakkara

---

## [Editor Report · Acceptance letter]

8 Nov 2024

PONE-D-24-07868R1 

PLOS ONE

Dear Dr. hongzeng, 

I'm pleased to inform you that your manuscript has been deemed suitable for publication in PLOS ONE. Congratulations! Your manuscript is now being handed over to our production team.

Kind regards, 

on behalf of

Dr. Pasyodun Koralage Buddhika Mahesh 

Academic Editor

PLOS ONE